# Cellular Mechanisms Participating in Brain Repair of Adult Zebrafish and Mammals after Injury

**DOI:** 10.3390/cells10020391

**Published:** 2021-02-14

**Authors:** Batoul Ghaddar, Luisa Lübke, David Couret, Sepand Rastegar, Nicolas Diotel

**Affiliations:** 1Université de La Réunion, INSERM, UMR 1188, Diabète athérothrombose Thérapies Réunion Océan Indien (DéTROI), 97400 Saint-Denis de La Réunion, France; batoul.ghaddar@univ-reunion.fr (B.G.); david.couret@chu-reunion.fr (D.C.); 2Institute of Biological and Chemical Systems-Biological Information Processing (IBCS-BIP), Karlsruhe Institute of Technology (KIT), Postfach 3640, 76021 Karlsruhe, Germany; luisa.luebke@kit.edu; 3CHU de La Réunion, 97400 Saint-Denis, France; 4CHU de La Réunion, 97410 Saint-Pierre, France

**Keywords:** adult neurogenesis, brain injury, neural stem cell, regeneration, stroke, zebrafish, mice

## Abstract

Adult neurogenesis is an evolutionary conserved process occurring in all vertebrates. However, striking differences are observed between the taxa, considering the number of neurogenic niches, the neural stem cell (NSC) identity, and brain plasticity under constitutive and injury-induced conditions. Zebrafish has become a popular model for the investigation of the molecular and cellular mechanisms involved in adult neurogenesis. Compared to mammals, the adult zebrafish displays a high number of neurogenic niches distributed throughout the brain. Furthermore, it exhibits a strong regenerative capacity without scar formation or any obvious disabilities. In this review, we will first discuss the similarities and differences regarding (i) the distribution of neurogenic niches in the brain of adult zebrafish and mammals (mainly mouse) and (ii) the nature of the neural stem cells within the main telencephalic niches. In the second part, we will describe the cascade of cellular events occurring after telencephalic injury in zebrafish and mouse. Our study clearly shows that most early events happening right after the brain injury are shared between zebrafish and mouse including cell death, microglia, and oligodendrocyte recruitment, as well as injury-induced neurogenesis. In mammals, one of the consequences following an injury is the formation of a glial scar that is persistent. This is not the case in zebrafish, which may be one of the main reasons that zebrafish display a higher regenerative capacity.

## 1. Introduction

Neurogenesis is an important process in which new neurons are formed from a pool of neural stem cells (NSCs). This process is initiated by the proliferation of NSCs leading then to the differentiation, migration, and the functional integration of newborn neurons into establishing and/or existing neuronal networks. Until recently, it was believed that neurogenesis only occurs during early embryonic development. However, Altman and Kaplan demonstrated in the 1960s and 1980s, respectively, that new neurons could also be produced in the brain of postnatal and adult rodents, as well as monkeys [1,2,3]. Since this pioneer discovery, an increasing number of works confirmed that indeed adult neurogenesis occurs in the brain of all vertebrates, including mammals [4,5,6]. Under physiological conditions, as well as after brain damage induced by traumatic brain injury (TBI), ischemia, or neuro-degeneration, NSCs play key roles in brain plasticity through the genesis of new neurons. Understanding the mechanisms regulating their activation and proliferation during regenerative and constitutive neurogenesis provides the chance to develop methods for combatting neurodegenerative diseases and disabilities following brain damage.

Adult neurogenesis is an important physiological process that supports brain plasticity and cognitive functions through the continuous generation of new neurons, allowed by the sustained activity of NSCs located in discrete brain regions called neurogenic niches. The persistence of functional neurogenesis during adulthood is evolutionary conserved from invertebrates (i.e., crustaceans, insects, etc.) to vertebrates including fish, amphibians, reptiles, birds, and mammals. However, the number of neurogenic niches, the proliferation rate of neural stem/progenitor cells, the migration, and differentiation of new neurons appears to differ according to species, brain size, and lifespan [6,7,8,9]. In mammals, the two main neurogenic niches correspond to the subventricular zone of the lateral ventricles (SVZ) and the subgranular zone (SGZ) of the dentate gyrus (DG) of the hippocampus. In striking contrast, the small teleost zebrafish (*Danio rerio*) displays a high number of neurogenic niches distributed throughout its entire encephalon. In addition, while regenerative neurogenesis is imperfect in mammals, teleost fish are able to repair their telencephalon from large injuries without any striking consequences and disabilities [9]. Such outstanding regenerative capacities strongly argue for a more comprehensive study of the molecular and cellular mechanisms allowing brain regeneration in teleost fish, in order to translate some important findings to humans.

In this review, we aimed at (i) describing the proliferative areas in the brain of fish and mammals, using mouse as an example; (ii) illustrating the spatial and cellular organization of the main telencephalic neurogenic niches in a comparative approach; and (iii) highlighting the similarities and differences regarding the spatiotemporal recruitment of the different cell types involved in brain repair (microglia, oligodendrocytes and their precursors, astrocytes, and NSCs). Concerning this last point, we will document the most studied models of brain damage: telencephalic mechanical injury in zebrafish and brain ischemia in mouse. Next, we will review the similarities and differences regarding neurogenic events and molecular mechanisms occurring after brain damage in zebrafish and mouse. Finally, we will highlight the value of zebrafish as a simple model for the analysis of brain repair mechanisms.

## 2. Location of Neurogenic Niches in the Brain of Adult Zebrafish, Rodents, and Humans

In the past, pioneer works using BrdU incorporation studies and/or Pcna (Proliferating Cell Nuclear Antigen) immunohistochemistry demonstrated the existence of areas with a strong proliferative activity along the ventricular/periventricular layers in the zebrafish brain [6,10,11,12,13]. In zebrafish, these strongly proliferative areas are widespread and can be detected throughout all the brain subdivisions including the telencephalon, the diencephalon, the mesencephalon, and the metencephalon (Figure 1A–C, left column) [11,12,14,15,16]. In the telencephalon of adult zebrafish, the main proliferative areas are located along the ventricle in the ventral, dorsal, dorsolateral, and posterolateral domains. Prominent domains of cell proliferation were also detected in the diencephalon, in the anterior and posterior parts of the preoptic area, as well as in the anterior, mediobasal, and caudal hypothalamus. In the posterior part of the encephalon, proliferation was also reported close to the rhombencephalic ventricle (Figure 1A–C). The thalamus, the regions surrounding the habenula, the pretectal periventricular region (a subdomain close to the optic tectum) and the three subdivisions of the cerebellum including the *valvula cerebelli*, the *corpus cerebelli*, and the *lobus caudalis cerebelli* all harbor substantial proliferation as well [6,11,14,17,18]. These proliferative regions are highlighted in red in a sagittal zebrafish brain section scheme, showing the distribution of neurogenic niches across the brain (Figure 1A).

In sharp contrast with zebrafish, there are only two main proliferative regions that have been observed in the brain of adult mammals: the SVZ of the lateral ventricles and the SGZ of the DG in the hippocampus [6,19] (Figure 1E,I). In addition to these two main regions, other discrete proliferative areas have been more recently observed in the brain of adult mammals, such as in the hypothalamus [20]. However, the number of proliferative cells in these domains remains lower than in the SVZ and SGZ. 

In both zebrafish and mammals, all these proliferative areas have been shown to generate a significant number of new neurons. Consequently, the adult zebrafish exhibits a strong neurogenic capacity due to the high number of active neurogenic niches throughout its brain, while adult mammals (rodents and human) display a limited number of neurogenic niches that are mainly localized in the SVZ and SGZ (Figure 1A,E,I) [6,10,11,14,21,22,23]. 

## 3. NSCs and Neural Progenitor Cells in the Adult Zebrafish and Mammalian Telencephalon

### 3.1. NSCs and Neural Progenitors in the Adult Zebrafish Telencephalon 

In zebrafish, the main neurogenic niches that have been studied during adulthood are located in the telencephalon, the optic tectum, and the cerebellum. The telencephalon remains undoubtedly the most investigated region of the brain, because it shares many features and homologies with the mammalian telencephalon, particularly considering adult neurogenesis [9,24,25,26]. In the telencephalon, several studies have explored the identity and the diversity of the neural/progenitor cells sustaining the strong neurogenic activity observed in the different telencephalic subdomains of the zebrafish brain [11,15,22,24,27,28]. In their initial work, Adolf and colleagues (2006) showed through BrdU incorporation studies and Pcna immunohistochemistry that the telencephalon contains two different types of neural progenitors: (1) slow cycling ones, distributed along the ventricular surface, and (2) fast cycling ones, organized mainly in a subpallial cluster [12] (Figure 2). The slow cycling progenitors were identified as radial glial cells (RGCs). In contrast, the fast-cycling cells were described as neuroblasts (Figure 1 and Figure 2) [11,14,15,28,29,30,31]. 

In the dorsal telencephalon, type 1 and type 2 cells correspond to quiescent and proliferative RGCs, respectively [15]. These cells are morphologically defined by a small triangular or ovoid soma localized close to the ventricle and extending two cytoplasmic processes: one short process towards the ventricular surface, and one long process crossing the brain parenchyma and reaching the pial surface. In mammals, RGCs were initially described as a scaffold for the migration of newborn neurons during embryonic neurogenesis, and were later shown to behave as NSCs [32], as in zebrafish [11,33]. At the end of the embryonic development in mammals, the majority of RGCs disappear by transforming into “conventional” astrocytes. However, RGCs persist during adulthood in the brain of adult zebrafish, maintain neurogenic properties, and support neuronal migration [9,11,34,35,36,37,38]. 

In adult zebrafish, these telencephalic RGCs were shown to perform symmetric and asymmetric division, and also, in some cases, to be able to directly convert into neurons [9,31,39]. In the pallium, lineage tracing and microscopy analyses showed that type 1 cells give rise to type 2 cells, which can give rise to type 3 cells (type 3 = neuroblasts) that are tightly inserted between RGC soma [25,31,39]. The newborn neurons will migrate radially along the long cytoplasmic RGC processes within the brain parenchyma to leave the ventricular zone [11]. At their target location they differentiate into mature neurons expressing well-characterized neuronal markers (i.e., HuC/D, Pax6a, PV) and display signs of functional integration such as synaptogenesis [11,16,31,40].

Adult RGCs in zebrafish express a set of well-identified markers (Table 1), including intermediate filaments (Gfap and Vimentin), the brain lipid binding protein (Blbp or fabp7), the calcium binding protein S100β, the estrogen-synthesizing enzyme (Aromatase B or Cyp19a1b), and also progenitor markers such as Nestin and Sox2 [9,11,15,28,29,37]. Recent studies also documented the expression of the inhibitor of DNA binding 1 (Id1), the chemokine receptor Cxcr4, *notch1a/b, notch3,* and *her4* genes in RGCs [40,41,42,43,44,45]. Most of these markers in zebrafish also label embryonic RGCs in mammals or neurogenic astrocytes during adulthood, as reviewed in [9]. 

Consequently, RGCs (type 1 and 2 cells) have been established as *bona fide* NSCs in the telencephalon of adult zebrafish [31,46], able to self-renew and to provide new neurons [9]. 

In the zebrafish telencephalon, type 3 cells correspond to fast-cycling progenitors and are considered as neuroblasts. These cells can be found tightly inserted between RGC soma in the pallium but are mainly localized within a subpallial cluster [15] (Figure 2). These progenitor cells undergo a limited amplification phase before performing symmetric neurogenic divisions [31,47]. Type 3 cells express committed progenitor markers such as *ascl1a* and PSA-NCAM (polysialylated neuronal cell adhesion molecule) in addition to progenitor markers, such as Nestin and Sox2 [15,48,49]. However, in general they do not express, or in some cases only weakly, the RGC markers. The type 3 cells can be divided into two subpopulations of neuroblasts: type 3a and type 3b (Table 1). Type 3a neuroblasts strongly express the commitment marker PSA-NCAM, but can also weakly express some of the RGC markers [15]. Type 3b neuroblasts do not express any RGC markers and are PSA-NCAM-positive [15]. Both types 3a and 3b express the Pcna proliferation marker.

The type 3 cells in the subpallial cluster will actively migrate, reaching the olfactory bulb via a rostral migratory stream-like (RMS) structure to differentiate into GABAergic and Tyrosine Hydroxylase-positive neurons. The zebrafish RMS-like is reminiscent of the mammalian RMS [16] (Figure 1B–D).

### 3.2. NSCs and Neural Progenitors in the Telencephalon of Adult Mammals

In mammals, the *bona fide* RGCs do not persist during adulthood [32,33], in contrast to zebrafish. However, in the SVZ and SGZ of the DG, RGCs transform into cells, which display astrocytic features and some of them maintain NSC properties during adulthood [6,19]. 

In the rodent SVZ, astrocyte-like cells (called B cells, *bone fide* stem cells) have been shown to self-renew and generate transit-amplifying cells (C cells) that give birth to neuroblasts (A cells) (Figure 1F,H) [50,51]. These neuroblasts will then migrate in chain following the RMS to reach the olfactory bulbs like in zebrafish. They will differentiate into GABAergic, glutamatergic, and dopaminergic neurons in the periglomerular layer of the olfactory bulbs and into GABAergic interneurons in the granular cell layer of the olfactory bulbs [19]. Interestingly, in the human SVZ, post-mortem studies have revealed that GFAP-positive astrocytes are separated from the ependymal wall by a hypocellular gap (Figure 1L). Only some of these astrocytes seem to proliferate and, therefore, the adult human SVZ appears devoid of newborn neurons that migrate in chain (no RMS). Supporting this notion, only very few new neurons displaying a migratory phenotype were observed in the anterior SVZ [52]. In contrast to rodents in which the newborn neurons from the SVZ migrate towards the olfactory bulb, they appear to migrate within the adjacent striatum to become medium spiny neurons in humans [53]. Consequently, the SVZ neurogenic niche differs greatly between humans and rodents in the cellular organization of the niche and in the newborn neuron migration [52]. 

In the rodent SGZ of the DG, radial glial-like/astrocyte cells (type 1) can self-renew and generate type 2a and 2b progenitors, which are also able to self-renew, and finally, type 3 cells (Figure 1G–L) [50,51]. These latter will give rise to glutamatergic dentate granule neurons. Interestingly, neurogenesis in the DG has been shown to be linked to learning, environmental enrichment and social interactions [19,54]. Newborn neurons in the SGZ migrate only short distances, in contrast to the new neurons from the SVZ that migrate for longer distances through the RMS [55]. Different studies in zebrafish suggest that the dorsolateral/dorsoposterior telencephalic domains could be the homologous of the SGZ in mammals, as recently reviewed in [9]. In the human hippocampus, the SGZ neurogenic niche is very similar to the rodent one. However, from a functional point of view, post-mortem studies supported the hypothesis that hippocampal neurogenesis strongly decreases during childhood to become almost undetectable at adulthood [56]. In contrast to that, in the same year, the work of Boldrini and collaborators showed through autopsy of hippocampi from healthy humans at different ages, that many immature neurons could be detected in the DG, suggesting that healthy older individuals maintain functional neurogenesis [4]. In conclusion, the hippocampal neurogenic niche shares many similarities with the one in rodents by locally generating neurons from neural precursors close to the niche [57] (Figure 1H,L). 

Taken together, these data demonstrate that zebrafish, rodents, and humans share similar features in the maintenance of adult neurogenesis with some homologies among the main telencephalic neurogenic niches, the type of NSCs, and neural progenitors as well as the type of newborn cells that are generated. Figure 1 highlights these similarities, as well as some differences between zebrafish, rodents, and humans. Interestingly, in humans, adult neurogenesis, or at least its functional relevance, is still under debate, especially when it comes to the neurogenic activity in the mammalian hippocampus [5,57,58,59].

## 4. Cellular Events Occurring after Telencephalic Injury in Zebrafish and Brain Damage in Mammals

Although the brain represents only 2% of the total body weight in humans, it consumes around 20% of the total body dioxygen and is highly active from a metabolic point of view using around 25% of the body’s glucose [60,61,62]. Thus, brain damage will strongly impact brain homeostasis through a decreased supply of nutrients (i.e., glucose and dioxygen) leading to severe outcomes. Better understanding brain plasticity could provide keys for combatting disabilities resulting from brain damage.

Teleost fish are widely used as a model for the investigation of brain plasticity due to their high constitutive neurogenesis, strong regenerative mechanisms, and striking sexual plasticity sustained by important sexual neurobehavioral changes [10,16,23,27,63,64,65]. Mechanical injury of the telencephalon by either inserting a small cannula through the skull or through the nasal cavity remains the most investigated model in zebrafish for studying brain regeneration [40,64,65,66]. Interestingly, the second injury method could lead to a damage of the olfactory bulbs, but does not alter the brain repair mechanisms substantially. 

After brain injury in teleost fish and mammals, death of damaged cells occurs, followed by the recruitment and/or proliferation of microglia and peripheral immune cells, oligodendrocytes/OPCs, astrocytes (only in mammals), endothelial cells, and NSCs. As part of the immune response, microglia can be activated and leukocytes can invade the injury site, both of which can release factors required for the activation and proliferation of NSCs, consequently leading to injury-induced neurogenesis. However, compared to zebrafish, mammals have a reduced ability to regenerate their brain and to fully recover sensory and motor functions. Understanding the cellular and molecular events occurring during brain regeneration is a challenging field of research but nevertheless important for the fight against disabilities resulting from brain damage. In the following sections, we will discuss the cascade of cellular events occurring after brain damage in zebrafish and rodents. 

Importantly, even if the injury models developed in zebrafish are closer to traumatic brain injury (TBI) models in mammals, we decided to mainly focus on stroke models in rodents because (i) the literature on stroke is much more abundant than TBI (PubMed research: “stroke rodent” → 31.199 articles versus “traumatic brain injury rodent” → 7.903, the 28 of January, 2021); (ii) TBI and stroke lead to almost similar cellular events and disorders such as gliosis, cognitive, neurological and psychological disorders [67]; (iii) TBI is a risk factor for stroke [68]. Furthermore, TBI and stroke share common molecular and cellular events including, among others, increased blood-brain barrier permeability, pro-inflammatory cytokine release, metabolic stress, glial reactivity, neuronal degeneration, axon damage, infarct formation, glial scar formation, nervous tissue atrophy and functional deficits [69]. Both pathologies also result in same recovery processed without striking differences in cognitive performance, suggesting similar regenerative outcomes [67]. The similar processes occurring in stroke and TBI are highlighted in Table 2. Last, but not least, it is difficult to have a realistic view of the cellular events occurring during brain damage due to the diversity of the TBI and stroke protocols. An integrative work has been realized showing cell death, astrocyte, oligodendrocyte and endothelium cell behavior from 3h post stroke to 1 week after a 30 min brain ischemia in mice [70]. This integrative work allows an easier comparison between brain repair mechanisms in fish and rodent and highlights the similarities and differences of brain reactivity following damage. For these reasons, we will mainly develop the comparison between stab wound injury of the telencephalon in fish and stroke in rodents, also discussing the respective cellular events occurring in the mammalian TBI model.

### 4.1. Cell Death after Zebrafish and Mammalian Telencephalic Damage

In zebrafish, very soon after mechanical injury of the telencephalon (from 4 h post lesion (hpl) to 6 hpl), numerous TUNEL-positive cells (Terminal deoxynucleotidyl transferase dUTP nick end labeling) are detected in both brain parenchyma and periventricular zone, while almost no cell death is observed in the contralateral control hemisphere [40,71]. The TUNEL-positive cells exhibit features of necrotic and apoptotic cells. In their work, Kroehne et al. showed that cell death could still be observed at 1 day post lesion (dpl) in both parenchymal and periventricular regions but returned to control levels at 3 dpl [40]. In contrast, Kyritsis et al. (2012) only observed a decreased number of TUNEL-positive cells at 3 dpl in the injured hemisphere when compared to the uninjured control hemisphere. In addition, the injury induced a strong edema, which represented 40% of the volume of the injured telencephalic hemisphere at 1 dpl [40]. At 7 dpl, this edema was strongly reduced to only 5% of the total volume of the injured hemisphere. Remarkably, 1 month after the injury, the lesioned hemisphere was almost completely restored regarding tissue morphology and histology. Moreover, no morphological differences could be observed anymore after 1 year [40]. An overview of cell death kinetics occurring after brain damage is shown in Figure 3.

In rodents, after brain ischemia, cell death progressively occurs within the first hours, as presented in zebrafish, but will persist for several days. This was shown through different methods using standard coloration, Fluoro–Jade C probes, and triphenyl tetrazolium chloride staining [72,73,74,75]. Additionally, numerous TUNEL-positive cells are detected after 1h, peaking at 24 h and can be still detected after 28 days in stroke models in rodents [76]. In rat, subjected to a 90 min ischemia by the Middle Cerebral Artery Occlusion (MCAO) method, the number of TUNEL-positive cells peaks at 48 h post-ischemia and returns to basal levels only 6 days post-stroke [77]. Several studies have shown that the processes/mechanisms promoting cell death are irreversibly initiated between 3 and 12 h post injury [78,79]. An overview of cell death kinetics occurring after stroke in mouse is shown in Figure 3. Considering TBI in mice, primary cell death occurs after injury followed by a second wave of neuronal cell death resulting from both biochemical and physiological disruptions, induced by the insult in a way similar to stroke [80,81]. 

Consequently, after brain damage, it seems that cell death is very severe and persists for several weeks in mammals including a secondary wave of neurodegeneration [82,83]. This differs greatly from the situation documented in zebrafish for which cell death is solved between 1 and 3 dpl. This important process of cell death occurring in mammals could trigger a chronic neuroinflammatory state that could be inhibitory for regenerative mechanisms.

### 4.2. Microglia Recruitment and Function in Response to Zebrafish and Mammalian Telencephalic Damage

Microglia are the resident immune cells of the central nervous system. In contrast to the other phagocytotic cells in mammals, microglia display strong interactions with neurons, astrocytes, and oligodendrocytes leading to a prominent role of microglia in neuronal development and plasticity [84,85]. As initially documented in mammals, the most striking characteristic of microglia is their high degree of plasticity, which enables them to switch from a resting state (quiescent) to a phagocytotic state (ameboid) in response to injury [86,87,88], a phenomenon also observed in zebrafish after telencephalic injury (Figure 4). After their activation, microglia cells start to secrete chemokines and attract leukocytes to the injury site. This process is then followed by phagocytosis where activated immune cells (including leukocytes) start to remove dying neurons, which helps to control inflammation and aids in tissue repair and functional recovery [89]. Activated microglia can release pro-inflammatory cytokines, including interleukins (IL-1β and IL-6) and the tumor necrosis factor (TNF-α) [90,91], as well as anti-inflammatory factors, such as TGF-β and the cytokines IL-4 and IL-10 [92,93,94], which are important for the different steps of brain repair.

In zebrafish, two microglial populations have been documented, differing in morphology, distribution, and functions [95]. The main population corresponds to phagocytotic microglia (ccl34b.1-positive) and is widely distributed, highly mobile, and phagocytic. The less represented microglia population (ccl34b.1-negative) is ramified and exhibits only low mobility and phagocytic properties [95]. As part of the inflammatory response, microglia appear to be among the first cells being recruited and activated following brain injury [61,66,91].

In zebrafish, performing L-plastin immunohistochemistry to label both microglia and leukocytes, an increasing number of L-plastin-positive cells can be observed from 6 hpl in the injured telencephalon, peaking at 24 hpl and decreasing slightly from 3 dpl to 5 dpl [71]. Accordingly, proliferative and non-proliferative microglial cells (ApoE-GFP or L-plastin positive) are shown to be largely increased at 3 and 4 dpl in the lesioned hemisphere compared to the unlesioned ones [40,66]. Taken together, these data demonstrate that in order to aid with brain recovery, microglial and potential peripheral immune cells are quickly recruited after brain injury, starting at 6 hpl before returning back to basal levels at 7 dpl. The general recruitment of microglia/immune cells after stab wound injury of the zebrafish telencephalon is shown in Figure 3. Recently, it was confirmed in zebrafish that microglia recruitment peaks at 1 dpl before it declines, remaining still significantly up-regulated at 4 dpl [96]. As in mammals, pro-inflammatory molecules (i.e., interleukins Il-1β, Il-6, Il-8, and TNF-α) secreted in part, by microglia have an impact on NSC plasticity, regeneration and neuronal repair, namely in injury-induced neurogenesis in zebrafish [71,96]. Although the exact contribution of microglia in brain repair mechanisms is still poorly understood in zebrafish, new data recently highlight it [96]. Indeed, the inhibition of microglia activation during zebrafish brain injury leads to a decreased expression of TNF-α and phospho-stat3/β-catenin signaling, which results in a lower proliferation of neural progenitor/stem cells and a lower number of newborn neurons without affecting differentiation [96]. These new data decipher the key roles of microglia in brain repair mechanisms. The phagocytotic activity of microglia and probably of other immune cells fortify the beneficial impact of inflammation on regeneration after telencephalic injury in the teleost fish. However, the precise function of these factors during zebrafish brain regeneration needs to be further investigated.

Similar to zebrafish, microglial cells in healthy brains of mammals remain stable and only a few of them are proliferating [97,98]. They are also among the first cells responding to brain injury/ischemia: they actively migrate to the injured site, switch from resting to ameboid states, and proliferate [99,100]. In mammals, microglial cells display a huge diversity of phenotype and reactivity allowing them different plasticity and functions. Indeed, microglia show different regional density (being more dense in the telencephalon and in myelinated regions), can be differently activated following injury (even in regions for which no neuronal death occurs) and can display morphological change with age [101]. As for macrophages, microglia can exhibit M1 (pro-inflammatory) or M2 (anti-inflammatory) phenotypes, the M2 phenotype being more associated with resolution of inflammation and regenerative processes, namely in stroke models [101,102,103]. In order to add complexity, it is also strongly suggested that microglial cells harbor different subtypes (at least 6) in the brain of mammals endowed with peculiar genomic, spatial, morphological, and functional specializations [104,105,106].

As nicely reviewed by Lourbopoulos and Benakis, within the first 24 h post stroke (hps), activated microglia are detected in both infarct and peri-infarct regions [107,108]. Between 2 and 7 days after stroke, microglia are further activated within the ischemic core [107,109]. Then, in the two following weeks, the number of microglia decreases in the peri-infarct and core regions. Interestingly, a substantial number of peripheral immune cells (neutrophils and macrophages) also invade the infarct and peri-infarct regions from day 1 post-stroke, due to the leakage of the blood–brain barrier and to chemoattractant factors. Their number is increased between day 3 and day 7 post-stroke but remains quite significant 7–14 days after ischemia [108]. Together, resident and peripheral immune cells play key roles in the removal of dead cell debris and potentially participate in limiting the damage to the surrounding nervous tissue. Of interest, microglia and macrophages will also accumulate around the damaged area, in a region where the glial scar will develop. Interestingly, it also appears that the sensitivity of microglial cells to brain ischemia is dependent on the regions [100]. The recruitment of microglia, during and after brain ischemia, is highlighted in Figure 3, and can be compared with zebrafish. Similar to stroke, TBI also induces microglia activation from 1 to 3 days post-injury, that can persist until 28 days after the trauma [110,111,112,113]. Other studies support these data with a significant increase in microglial cells 2 and 14 days post-TBI, as well as a microglia shape that remains different at 60 days post-TBI compared to their control phenotype [114].

Interestingly, microglia have also been shown to be part of the neurogenic niches and to produce positive and negative effects on neurogenesis according to their activation state and panels of secreted molecules. Thus, ischemia and cell death could initiate IGF-1 and TGF- β expression and subsequently promote neurogenesis [101]. Very interestingly, new data strongly suggest that the resident microglial population does not inhibit endogenous brain regeneration processes in mouse following TBI, but rather cannot support these processes [115]. A pro-regenerative phenotype can also be induced in mammals through IL-6 trans-signaling [115], demonstrating that inflammatory signals are important contributors to brain repair mechanisms as in regenerative organisms like zebrafish.

Consequently, compared to zebrafish, microglial recruitment and activation is prolonged in mammals, which is possibly linked to the persistent cell death occurring within the damaged hemisphere. Such a persistent cell death, as well as microglia/peripheral immune cells recruitment could induce chronic disruption of brain homeostasis, impairing consequent brain repair mechanisms.

### 4.3. Oligodendrocyte/Oligodendrocyte Progenitor Cell Recruitment after Zebrafish and Mammalian Telencephalic Damage

Oligodendrocytes are among the most important cells within the central nervous system as they participate in the development and maintenance of the myelin sheath. In mammals, mature oligodendrocytes lose their proliferative capacity and newly generated oligodendrocytes derived from non-myelinated oligodendrocyte precursor cells (OPCs) [116].

To investigate the recruitment of oligodendrocytes and OPCs in the zebrafish brain, März et al., (2011) used an *olig2*:EGFP transgenic line. They observed an increased number of OPCs and mature oligodendrocytes at 1 dpl (in 50% of the studied brains). This accumulation of OPCs is more prominent between 2 dpl and 14 dpl and is detected in almost all the studied brains (94%). Interestingly, at 35 dpl, the *olig2*:EGFP clusters are almost not observed anymore in the injured hemisphere (März et al., 2011). Surprisingly, in contrast to mammals, there is no increase in the proliferation rate of olig2-positive cells in the injured hemisphere, compared to the uninjured hemisphere (März et al., 2011, Baumgart et al., 2012). However, recent data suggest a higher number of proliferating parenchymal and ventricular olig2-positive cells at 4 dpl [117]. This study also suggests that olig2-positive RGCs from the medial telencephalic ventricular zone can generate new oligodendrocytes following brain injury [117]. Consequently, the proliferation of the olig2-positive cells appears to be moderate after stab wound injury of the zebrafish telencephalon. In summary, immune cells (microglia and peripheral cells) and oligodendrocytes/OPCs are among the first cells recruited and activated after stab wound injury in zebrafish. At 2 dpl, around 50% of the proliferative cells within the damaged brain parenchyma could be identified as endothelial cells, olig2-GFP positive cells, and microglial-like/immune cells [40]. The general recruitment of olig2-positive cells after injury is shown in Figure 3.

In mammals, oligodendrocytes are sensitive to cerebral ischemia and TBI [118,119,120,121], and their death, as well as the loss of the myelin sheath strongly impairs neuronal function. After brain ischemia, lineage tracing showed that OPCs are generated from NSCs located in the SVZ, and provide new oligodendrocytes [120,122,123]. Thus, a significant increase in OPCs is observed, giving rise to mature myelinating oligodendrocytes in the peri-infarct gray and white matter where sprouting axons are located [120,124,125,126]. This oligodendrogenesis has been shown to improve brain repair processes and neurological scores [120]. After brain ischemia, OPCs also seem to be involved in post-stroke angiogenesis [127], a process linked to neurogenesis [128]. Indeed, OPCs in the cerebral cortex shift from a parenchymal to a perivascular subtype. The recruitment of oligodendrocytes/OPCs during and after brain ischemia is highlighted in Figure 3. In the TBI model, mature oligodendrocytes undergo apoptosis occurring from 2 days to 2 weeks after the insult. In parallel, olig2-positive cell proliferation is observed starting at 48 h and can persist until 21 days after the injury [121,129]. These data show that OPCs respond to brain injury in a way similar to what was shown for stroke. Such a proliferation may lead to the genesis of new oligodendrocytes contributing to remyelination.

Consequently, the situation is very different in mammals compared to zebrafish, as the number of OPCs is significantly increased, and they actively proliferate providing numerous new oligodendrocytes. In zebrafish, the proliferation rate of olig2-positive cells remains low and their number is unchanged during the regenerative process, although olig2 clusters are observed in close vicinity to the lesion [66]. Therefore, oligodendrogenesis appears to be vastly different between zebrafish and rodents. The role of olig2-positive cells at the lesion site remains largely unknown in zebrafish, but could be linked with regenerative neurogenesis, axonogenesis, and synaptogenesis.

### 4.4. Injury-Induced Proliferation and Neurogenesis after Telencephalic Damage in Zebrafish and Mammals

After zebrafish telencephalic lesion, brain cell proliferation occurs with different kinetics within the brain parenchyma and in neurogenic niches [65]. Simultaneous to the recruitment of immune cells and the accumulation of OPCs, starting between 1 and 2 dpl, a higher number of Pcna-positive cells can be detected in the injured hemisphere. After 48 hpl, this proliferation is especially observed along the ventricular layer where RGCs reside. This number peaks between 5 and 8 dpl and slowly decreases until 15 dpl to reach the normal proliferation rate at 35 dpl [10,64,65,66,130,131]. Double immunohistochemistry against proliferation and RGC markers, as well as the use of transgenic fish, have shown that the reactive proliferative cells localized along the ventricle correspond to RGCs expressing S100β, Blbp, Gfap, and also Vimentin [40,64,65,66]. This injury-induced proliferation of RGCs has been shown to produce newborn neurons (HuC/D-, parvalbumin-positive), which persist for more than 2 to 3 months after brain injury [40,64]. They also exhibit the MAP2(a+b) dendritic marker, the synaptic vesicle marker SV2 and the synaptic marker metabotropic glutamate receptor 2 (mGlu2), proving their functional maturation [40]. Consequently, after telencephalic injury, RGCs switch from a quiescent to a proliferative state and generate newborn neurons to replace neurons, which have been lost due to the damage. Interestingly, compared to constitutive neurogenesis, a shift in the mode of division of NSCs has been observed by Barbosa and colleagues during regenerative neurogenesis [46], which could lead to a depletion of NSC.

An important aspect to consider in injury-induced NSC proliferation is the influence of the pro-inflammatory cytokines transiently upregulated after telencephalic injury and shown to be necessary for NSC activation [71]. Furthermore, the transcription factor Gata3 is required for reactive proliferation of RGCs and the subsequent regenerative neurogenesis [131]. Interestingly, the Gata3 transcription factor is mainly expressed by RGCs but is also detected in L-plastin positive microglia [131], pointing again to an important role of the inflammatory response (leukotriene/gata 3 [131]). In the same line of evidence, the new data obtained demonstrate the role of microglia activation and TNF-α in regenerative neurogenesis in zebrafish [96].

Considering injury-induced neurogenesis in mammals, newborn neurons are produced from the SVZ and migrate within the injured striatum and cortex of rodents after stroke. During their migration, they will progressively differentiate and express neuronal markers (i.e., DCX (doublecortin), PSA-NCAM, Hu, and NeuN) [132,133,134]. Interestingly, these newborn neurons do not reach the olfactory bulbs through the RMS, as during constitutive neurogenesis, but reach the damaged areas due to attractive factors [134,135]. These new neurons are highly detectable between 14 and 28 days after stroke [134]. Consequently, after stroke, NSCs from the SVZ give rise to neuroblasts that migrate towards the damaged regions (striatum and cortex), where they differentiate into mature neurons [135]. New migrating neuroblasts can still be observed 1 year after stroke [136]. In addition, after stroke, hippocampal neurogenesis is also detected, but remains imperfect [137]. In rodents, TBI models also display injury-induced neurogenesis [138,139,140]. Thus, TBI has been shown to promote the reactivation of quiescent NSCs that actively divide producing new neural progenitors (Wang et al., 2016). The newly generated neuroblasts will migrate in chain to the lesioned areas (Chang et al., 2016). As reviewed by Chang et al. (2016), the different models of TBI seem to globally lead to increased NSC proliferation, migration, and differentiation, but a wide heterogeneity in the TBI responses is observed probably due to the differences between the severity, location, timing, and types of injury.

The similarities and differences regarding cell activation and cell recruitment after brain injuries in mammals and zebrafish are highlighted in Table 3.

### 4.5. Reactive Astrogliosis after Brain Injury

After any type of brain damage in mammals (i.e., stroke, TBI, neurotoxic drug exposure, neurodegenerative disease), astrocytes surrounding the damaged region will react and undergo important morphological and/or functional changes, such as hypertrophy, overexpression of some genes or astrocytic markers, such as GFAP and Nestin, which will progressively modify their function [78,141,142,143,144]. Although all the astrocytes surrounding the damaged area react, they do not constitute a homogenous population; at least two different types of reactive astrocytes have been described. The astrocytes in close vicinity to the lesioned site will start to proliferate and migrate surrounding the injured territory. These astrocytes are of peculiar importance for the establishment of the well-known glial scar (mainly composed of extracellular matrix and numerous processes from astrocytes). The astrocytes that are further away from the lesion site will also react but will stay resident and maintain their connection to the neighboring cells.

Under stroke conditions, proliferation of astrocytes starts in the first days and remains restricted to an area 200 micrometers around the infarcted site [145,146]. Interestingly, the inhibition of astrocyte proliferation increases the size of the injury and worsens neurological scores, correlated with a higher neutrophil infiltration and impaired blood–brain barrier regeneration, as shown for TBI [147,148,149]. New astrocytes are also generated from NSCs that could migrate from the SVZ a few days after stroke onset, and could survive until several weeks after the stroke [150].

Reactive astrocytes become hypertrophic with thicker and bushier/ramified processes; they also upregulate many astrocytic markers, such as GFAP [151]. GFAP upregulation after brain ischemia will be weak at 24h post-injury but will increase rapidly during the first week [152,153], while the number of GFAP-positive astrocytes increases within the first two weeks [154,155]. Vimentin and Nestin, two other intermediate filaments are also upregulated after ischemia and their expression levels correlate with those of GFAP [152,156]. Interestingly, a single knock-out of GFAP or Vimentin has no real impact on reactive gliosis and glial scar formation while a double knockout (KO) severely impacts reactive gliosis, as shown by a decrease in astrocyte hypertrophia and in glial scar formation. Under stroke conditions, the double KO of these intermediate filaments increases the size of the infarct and leads to more acute neurological outcomes [157,158]. Similarly, in TBI models, astrogliosis also occurs through structural and functional changes including hypertrophy and overexpression of intermediate filaments (Nestin, Vimentin, and GFAP) [159,160]. Astrocytes appear as key cell in the development of the glial scar that is essential in the establishment of a physical and chemical barrier that isolates the damaged area and contains the spread of inflammatory cells. Inhibiting or promoting astrogliosis (and so glial scar) did not have striking curative effects [159]. However, the selective stimulation of beneficial astrocyte-derived molecules could represent an interesting therapeutic way to promote blood–brain barrier repair, neurogenesis, and synaptic plasticity [159].

Remarkably, as there are almost no astrocytic cell-like structures in the brain of adult zebrafish, no astrogliosis occurs following injury in zebrafish. Nevertheless, RGCs that express markers of mammalian astrocytes such as Gfap, Nestin, or Vimentin, also up-regulate these markers after telencephalic damage endowed with RGC process hypertrophy [64]. In addition, RGCs also proliferate as previously discussed. This feature constitutes a major difference between zebrafish and mammals and may have strong implications in the regenerative processes particularly concerning the glia scar process.

### 4.6. Glial Scar: A Paradigm for Understanding the Difference between Zebrafish and Mammalian Regeneration?

After brain damage in mammals, reactive gliosis takes place involving microglia, oligodendrocyte, and astrocyte cells. Activation of astrocytes will lead to the formation of the glial scar. The glial scar is supposed to protect the central nervous system and to participate in the healing process by forming a physical and chemical barrier that isolates the damaged area and contains the spread of inflammatory cells [161]. Thus, inhibiting the glia scar formation during brain injury has been shown to worsen damage [157,158]. During the glial scar formation, reactive astrocytes secrete many extracellular matrix components such as laminin, fibronectin, tenascin C, and proteoglycans [141,162]. In addition to representing a physical barrier, these extracellular matrix molecules can also lead to growth cone collapse, axonal guidance inhibition, as well as neural progenitor migration defects through activation of RhoA/ROCK signaling [163]. In zebrafish, as no astrocyte-like structures were observed in the telencephalon, RGCs are suggested to sustain many astrocytic features and functions, such as typical marker expression, steroidogenesis, blood-brain barrier establishment, and neurogenic properties [26,164]. Although, no astrogliosis is observed in zebrafish after telencephalic injury, reactive RGC gliosis occurs, as shown by the up-regulation of Gfap and Vimentin, as well as the hypertrophy of RGC glial processes [40]. The upregulation of RGC markers (Vimentin, Gfap, Blbp, and S100β) and the hypertrophy of RGC processes is observed quickly after brain damage and can remain visible up to 1 month after injury. In addition, numerous studies also demonstrate an increase in RGC number following brain injury in the telencephalon [37,41,60,61,125]. Interestingly, collagen acid-fuchsin-orange G staining confirms the transient accumulation of collagen at 14 dpl at the injury site [40]. However, this fibrotic scar formation, including reactive glial cell accumulation, hypertrophy of glial processes, persistence of inflammatory cells and ectopic extracellular matrix deposition are not detected later or just occasionally in a small number of brains in zebrafish. Remarkably, the work from Baumgart and colleagues reports that RGC hypertrophy and reactivity is observed for large lesions but not for small ones. Therefore, RGC reactivity could be linked to the severity of the damage [64], as astrogliosis in mammals. Furthermore, discrete lesions will only allow the proliferation of RGCs, while larger lesions could potentially initiate the additional migration of some RGCs within the brain parenchyma. However, such migration should remain quite discrete, as the analysis of stab wounded brain sections did not demonstrate any migration processes in the past.

Consequently, it appears that RGC reactivity mimics, in part, astrogliosis with respect to the mammalian situation through (1) increased expression of glial markers; (2) increased proliferation and hypertrophy; (3) potential migration of RGCs in some cases; and (4) the increased extracellular matrix deposition. However, unlike in mammals, there is no evidence for permanent scar formation in the zebrafish brain as there is no persistent extracellular matrix deposition. An overview of proliferation and RGC reactivity occurring during brain lesion is shown in Figure 3.

### 4.7. Brain Damage: What about Humans?

When it comes to the close investigation of the consequences of brain damage, for example due to stroke or TBI, unfortunately in humans, studies are highly limited due to the incapacity of collecting post mortem tissue after the onset of damage. However, it was shown that apoptosis occurs quickly in the human brain after ischemia with cell death being delayed for several days [165,166]. Similar to the situation in rodents, microglia are recruited and proliferate at the periphery of the damaged area in post-mortem human brain tissue of stroke patients [167], and peripheral immune cells are attracted as well [168,169]. Furthermore, in the peri-infarct region after ischemic stroke in humans, the development of a glial scar can be observed [170], as well as injury-induced neurogenesis [135,171].

## 5. Conclusions

Brain ischemia, traumatic brain injuries, and neurodegenerative diseases are of major concerns worldwide and constitute main health issues. These brain damages can lead to severe disabilities, including cognitive, sensorimotor, and even personality dysfunctions. Unfortunately, the brain plasticity and regenerative capacities of mammals are strongly limited. The blunted regeneration observed in mammals is largely attributed to inflammatory processes inducing the formation of a glial scar. For these reasons, the study of highly regenerative species is important for cross-comparison and for a better understanding of the molecular and cellular mechanisms that enable these organisms to efficiently regenerate without displaying disabilities [172]. Several hypotheses have been advanced in order to explain the strong plasticity of zebrafish. Among them, zebrafish appear to respond to brain damage by turning on genes, such as *gata3* and *interleukin-4* receptor that are not activated in rodent models [173,174,175,176]. Additionally, changes in regeneration-responsive enhancers of mammals might be another reason for less regenerative capacities in mammals, when compared to zebrafish. This was recently suggested for the *inhibin beta A* gene [177]. Furthermore, they display an immune response allowing an enhanced regeneration [96,178]. Last, but not least, zebrafish are still growing during their entire lifespan, and their brain seems to retain some embryonic features that could explain, in part, their strong regenerative capacities [10].

One first interesting aspect to be considered is that neural stem/progenitor cells react similarly by increasing their proliferation after brain injuries in zebrafish and mammals. In both taxa, proliferation of neural stem/progenitor cells peaks at around 7 days post injury. However, the vast majority of freshly generated newborn neurons after injury in mammals fails to reach the damaged site due to the formation of the glial scar. This scar gliosis provides an extracellular environment that does not allow the integration of newborn cells and is a potential inhibitor of neurogenesis (i.e., chondroitin sulfate proteoglycans and myelin components) [179]. As a result, new neurons are unable to cross the glial scar, will degenerate and can therefore not compensate the functions lost during the massive neuronal death induced by brain ischemia or injury. In contrast, although brain lesion in zebrafish strongly induces the proliferation of NSCs, as in mammals, it will not lead to the formation of a strong and persistent glial scar. This will consequently allow the migration of new neurons to the injured site and lead to their functional integration and to the recovery of impaired functions. These data comfort the general idea that the glial scar has a negative impact on newborn neuron integration in the brain of mammals. Data suggest that considering the glial scar as good or bad for CNS recovery is not as simple as suggested [180,181]. In spinal cord injury, the initial glial scar formation is thought to limit the spread of inflammation, but secondly impairs spinal cord regeneration [182]. However, the temporal targeted and moderated modulation of the glial scar formation could represent an interesting way to promote brain recovery, such as the case for the spinal cord.

Another interesting aspect is that the overall reactivity of the brain, following brain damage in mammals and zebrafish, is quite similar. It involves the death of parenchymal and periventricular cells (neurons and glia) that will lead to the recruitment and the activation of glial cells. However, cell death and microglia reactivity will persist longer in the brain of mammals than in zebrafish. These processes may lead to a chronic neuroinflammation that does not exist in the brain of fish, promoting an inflammatory microglia state and avoiding neuronal replacement. In contrast, the positive role of microglia in zebrafish telencephalic regeneration has been recently highlighted [96,172]. Importantly, in contrast to mammalian microglia cells, for which recent transcriptomic analysis have shown a relative homogeneity during adulthood, zebrafish microglia can be separated into two subpopulations (ccl34b positive and negative) displaying different phenotypes and functions [95]. Wu et al. raised the question of the functional specialization of microglial in fish linked to multiple rounds of genomic duplication [183]. Moreover, during zebrafish brain regeneration, microglia display a M2 (anti-inflammatory) phenotype that is known to favor tissue repair in mammals [90]. Interestingly, modulation of the microglia phenotype in mice was shown to promote newborn neuron survival and cognitive function in a TBI model [115]. Consequently, the modulation of microglia activation and functions could be a major element for improving brain recovery.

Of note, another important event to consider when discussing brain recovery is angiogenesis. In addition to serving as an important scaffold for neuroblast migration, blood vessels can secrete important factors (i.e., prostacyclin) promoting axonal growth and subsequent recovery [179]. Actually, it is more and more admitted that improving angiogenesis could favor neurogenesis and brain recovery [128]. Thus, neurogenesis cannot be considered anymore as the only way to improve functional recovery from stroke or brain damage, but should be examined in connection with angiogenesis, as it seems that angiogenesis and neurogenesis are coupled [184].

Finally, from an evolutionary point of view, a remaining question is whether the mammalian brain lost its regenerative capacities or if it inhibits the regenerative capacities. Another possibility is that the teleost fish have developed such capabilities independently of mammals. One of the main differences between these taxa remains that, in zebrafish, NSCs retain a part of their embryonic features allowing their high reactivity and plasticity and continuing growth of the brain during adulthood [10]. Last, but not least, a comprehensive understanding of the mechanisms by which the glial scar is transiently generated and resolved in zebrafish could open a way for promoting brain regeneration in mammals and avoiding the consequences of brain damage. In addition, it could also be argued that mammals lost the ability to drive the expression of key genes involved in the regenerative process, due to a major regulatory change in their expression following injury [177]. Interestingly, studies on invertebrates such as drosophila also demonstrate common strategies in neurogenesis and brain repair with mammals and highlight the role of blood vessels in these mechanisms [185,186]. Thus, the study of different taxa is of great interest in order to better understand neurogenesis and brain repair through evolutionary conserved processes.

In summary, it seems that proposing multifactorial therapeutic approaches targeting cell death, microglia, OPCs, astrocytes, and NSCs could be more efficient for improving regeneration than targeting only one mechanism.

## Figures and Tables

**Figure 1 cells-10-00391-f001:**
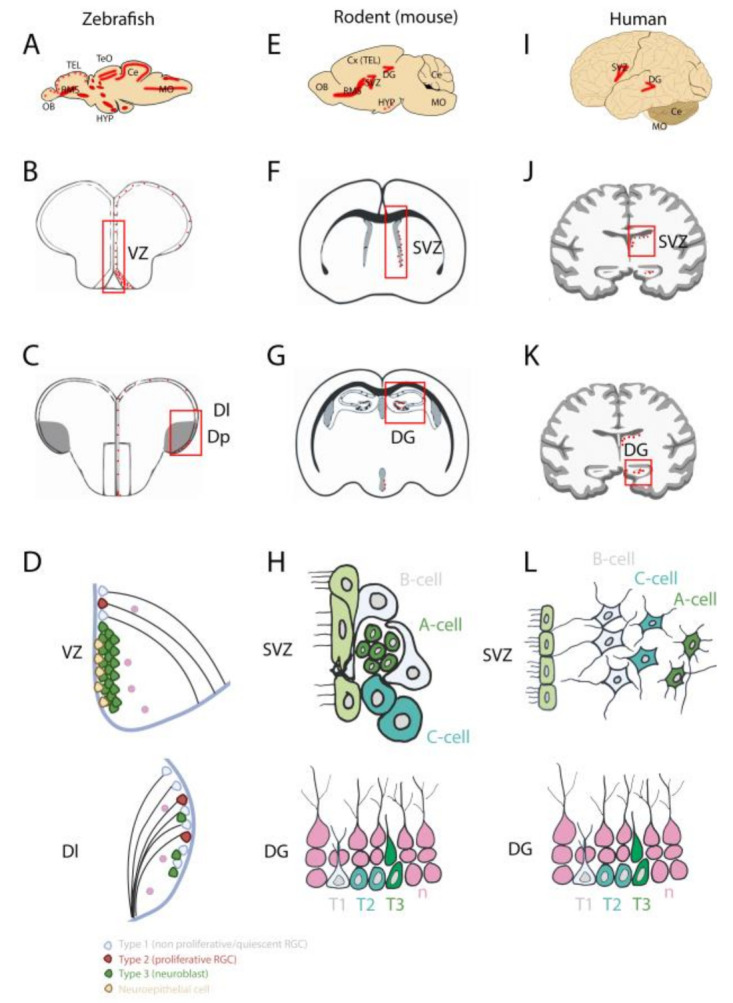
Localization and cellular organization of the main neurogenic niches in the brain of adult zebrafish, mouse, and humans. (**A**,**E**,**I**): sagittal sections of zebrafish (**A**), mouse (**E**) and human (**I**) brains with the main proliferative regions (neurogenic niches) shown in red. The mammalian brain displays only two main neurogenic niches: the subventricular zone (SVZ) of the lateral ventricles and subgranular zone of the dentate gyrus (DG) of the hippocampus. Note that the mammalian hypothalamus (HYP) also exhibits discrete neurogenesis. The zebrafish brain displays numerous niches throughout the brain. (**B**–**K**): transversal sections through the brain, marking the main neurogenic niches of the respective species shown in (**A**,**E**,**I**). (**D**–**L**): Cell composition of the neurogenic niches in zebrafish, mice and humans. (**D**): The main neurogenic niches in the subpallial ventricular zone (VZ), the dorsolateral telencephalon (Dl) in zebrafish, and their respective homologues in mammals: the SVZ and the DG of the hippocampus in mouse and humans. In zebrafish, type 1 and type 2 cells are quiescent and proliferative radial glial cells (RGC), respectively (quiescent and proliferative neural stem cells (NSCs)). Type 3 cells are proliferative neuroblasts. The neuroepithelial cells are NSCs from the subpallium. (**H**,**L**): In mammals, the NSCs are shown in grey (B-cells and Type 1 -T1-), the transient amplifying cells in light green (C-Cells and Type -T2-) and the neuroblasts in dark green (A-cells and Type 3 -T3-). Note the hypocellular gap in the human SVZ compared to mice. Ce: cerebellum; Cx: cerebral cortex; Dl: lateral zone of the dorsal telencephalic area; DG: dentate gyrus of the hippocampus; Dp: posterior zone of dorsal telencephalic area; HYP: hypothalamus; MO: medulla oblongata; OB: Olfactory bulbs; RGC: radial glial cell; RMS: rostral migratory stream; SVZ: subventricular zone VZ: ventricular zone; TEL: telencephalon; TeO: optic tectum.

**Figure 2 cells-10-00391-f002:**
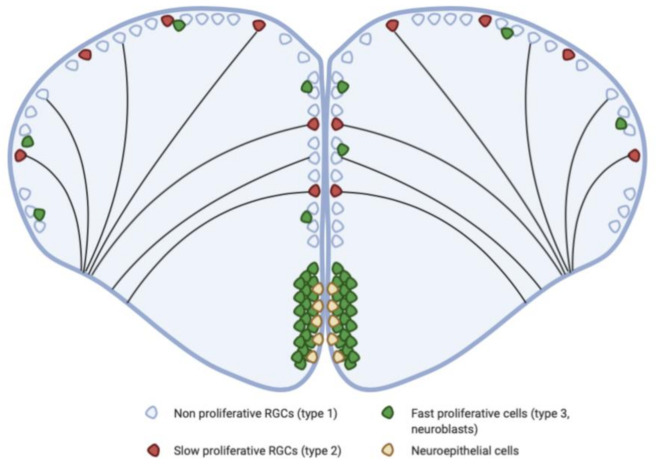
The telencephalon of adult zebrafish contains slow and fast cycling progenitors. The VZ of the dorsal telencephalon (pallium) is mainly composed of quiescent (type 1) or proliferative (type 2) RGCs corresponding to slow cycling progenitors. The ventral part of the telencephalon (subpallium) is composed of fast cycling progenitors (type 3 cells) identified as neuroblasts, grouped within a cluster and forming a rostral migratory like structure (RMS-like). Some neuroblasts are also observed scattered between RGC soma in the pallium. RGCs were identified as bona fide NSCs in the pallium and neuroepithelial cells could be NSCs in the subpallium. RGCs: radial glial cells.

**Figure 3 cells-10-00391-f003:**
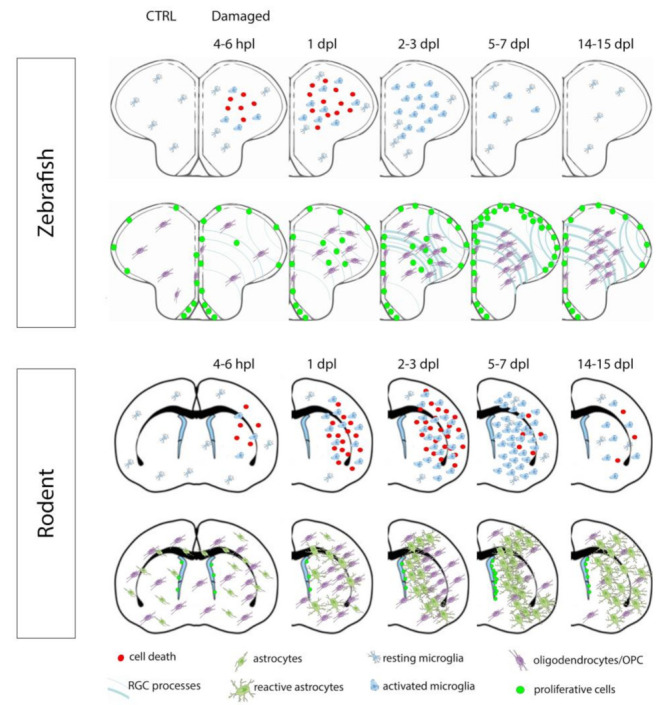
Cellular events occurring after telencephalic injury in zebrafish and stroke in mouse. After brain damage, numerous cells, mainly neurons, die. This process is followed by the activation and recruitment of microglial and other immune cells (leukocytes) in parallel to OPCs. Then an astrogliosis process occurs in mice, while RGCs become reactive and proliferative in zebrafish. Proliferation in the neurogenic niches peak at day 7 after damage in both models. Note that the cellular response on the contralateral side is not shown for zebrafish and rodent. In zebrafish and rodent, the first row shows cell death, microglial recruitment, and activation. In zebrafish, the second-row highlights the hypertrophy of RGC processes, the neurogenic injury-induced proliferation as well oligodendrocytes/OPCs response. In rodents, the second row shows astrogliosis, neurogenic proliferation along the SVZ and oligodendrocytes/OPCs response. Note that in zebrafish oligodendrocytes/OPCs accumulate close to the lesion site without increasing their number; dpl: day(s) post lesion, hpl: hour(s) post lesion.

**Figure 4 cells-10-00391-f004:**
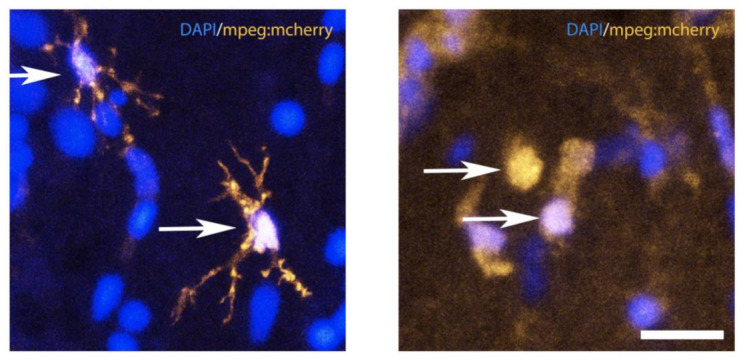
Resting and activated microglia under injured and uninjured (control) conditions in the telencephalon of zebrafish. Confocal microscopy showing quiescent (resting) microglia (left panel) and ameboid (activated) microglia (right panel) in the adult zebrafish telencephalon. There is an obvious change in the shape of the microglia between injured and uninjured tissue, illustrated by the *mpeg*:mcherry transgenic fish line, which labels microglia in the central nervous system. Arrows show the resting morphology of microglia cells (left panel) the ameboid shape of activated microglia at 1dpl (right panel). Bar: 18 μm.

**Table 1 cells-10-00391-t001:** Main markers expressed by type 1, 2, 3a, and 3b cells in the telencephalon of adult zebrafish. The (+/−) means that these markers are expressed at lower levels in the subtype.

Type 1	Type 2	Type 3a	Type 3b
Sox2	Sox2	Sox2	Sox2
Nestin and Vimentin	Nestin and Vimentin	Nestin	Pcna
Gfap	Gfap	S100 beta (+/−)	PSA-NCAM
S100 beta	S100 beta	Blbp (+/−)	
GS (glutamine synthetase)	GS	AroB (+/−)	
Blbp	Blbp	Pcna	
AroB	AroB	PSA-NCAM	
Cxcr4	Cxcr4 (+/−)		
Id1	Id1 (+/−)		
Her 4	Her 4		
	Pcna		

**Table 2 cells-10-00391-t002:** Similarities of processes occurring in stroke and traumatic brain injury (TBI) in mammals. +: present; −: absent (adapted in part from the work of [69]).

	Stroke	TBI
Blood-brain barrier permeability	+	+
Metabolic stress/Ionic perturbation/Cytokine	+	+
Membrane damage/Contusion/Primary axotomy	−	+
Glial swelling/Blood flow reduction/Inflammation/Secondary axotomy	+	+
Cell death and Wallerian degeneration	+	+
Infarct formation	+	+
Nervous tissue atrophy	+	+
Cognitive and sensorimotor deficits	+	+
Reactive gliosis (microglia, astrocyte, oligodendrocytes)	+	+
Glial scar	+	+

**Table 3 cells-10-00391-t003:** Comparison of events after brain damage in zebrafish and mammals. +++: strong; +: present; +/−: weak; −: absent.

	Zebrafish	Mammals
Glia reactivity/hypertrophy	+	+
Microglia recruitment	+	+
Microglia proliferation	+	+
Oligodendrocytes recruitment	+	+
Oligodendrocytes proliferation	+/−	+
Astrocyte/RGC recruitment	− (RGC)	+ (astrocyte)
Astrocyte/RGC proliferation	+ (RGC)	+ (astrocyte)
GFAP/vimentin up-regulation	+	+
Glial scar formation	−	+
Glial scar persistence	−	+
Regenerative capacities	+++	+/−

## Data Availability

Not applicable.

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
