# Peer review of "Cellular Mechanisms Participating in Brain Repair of Adult Zebrafish and Mammals after Injury"

_cells, 2021, doi:10.3390/cells10020391_

Round 1
Reviewer 1 Report
Review Report
Title: Cellular mechanisms participating in brain repair of adult zebrafish and mammals after injury
This review presents two interesting topics on brain repair: adult neurogenesis and cellular mechanisms triggered by injury, focusing both on similarities and differences in these processes in adult zebrafish and mammals. The authors selected an impressive amount of impactful literature in the area and outlined important aspects related to brain repair in evolutionary and physiological contexts. However, its presentation suffers from having a confusing structural organization, as well as being mostly descriptive with less thoughtful discussion/comparison between zebrafish and mammals. Providing more details of the material across species would significantly enhance the value of this review.
General comments
- The abbreviations need to be carefully reviewed all over the manuscript.
- The structural organization of the manuscript is a bit confusing. In the abstract, the authors state that they “will first discuss the similarities and differences regarding (i) the distribution of neurogenic niches in the brain of adult zebrafish and mammals (mainly mouse)” but this was not clear until page 12-13 when they finally describe the characterization of rodent neurogenic niche and then show the similarities and differences. The same happens with other topics, such as brain damage and brain repair. Rather than presenting this information in this linear fashion, it would be more interesting and helpful for the reader to integrate the content related to the different animal models.
- It’s not clear why authors focused on TBI models in zebrafish and stroke models in rodents to compare cellular events after brain injury. Although “TBI and stroke lead to almost similar cellular events and disorders – as stated by the authors -, these injuries have different types of primary insult and then different cellular vulnerability, which becomes a challenge to study timeline events related to inflammatory responses. It would be more interesting to focus on TBI models in rodents as well and use the stroke data (as a secondary topic) to reinforce the main idea on brain injury.
- The discussion is slightly superficial. Some topics are missing detail and focus.
Specific comments
Line 18 (Abstract). “we will first discuss the similarities and differences regarding (i) the distribution of neurogenic niches in the brain of adult zebrafish and mammals (mainly mouse)”. These similarities and differences are not clear at the beginning of the manuscript, but instead wait several pages for the comparison information to appear. It would be far more helpful for the reader if the authors could use their expertise to integrate this information and compare these models from the beginning, as is suggested by the abstract.
Page 4 (Figure 1). This is a great figure that brings together a nice comparison of neurogenic niches throughout the brain in the three groups (zebrafish, rodents, and humans) but the discussion of similarities and differences is spread over the text. For example, page 2 (line 77) describes the neurogenic niche in zebrafish but the rodent’s characterization is only described on page 12 (line 355). Why not put this information together? The same happens in Figure 3.
Line 105. Please check the legend of Figure 1; it is missing a couple of abbreviations, such as RGC, Dp, Cx, and RMS. Also, consider adding letters to Figure 1 (ex., E, F, G…) to identify rodent and human brain sections.
Line 107. “The mammalian brain displays only two main neurogenic niches: the subventricular zone (SVZ) of the lateral ventricles and subgranular zone of the dentate gyrus (DG) of the hippocampus.” Figure 1 shows a neurogenic area in the hypothalamus (Figure 1A) of rodents as well; it would be helpful to mention that in the legend or consider removing it.
Figure 1, 2, and line 141. The nomenclature of RGC cells is not consistent. Is there any difference between “Non proliferative RGCs (Type1)” or “quiescent (Type 1)” or “Type 1 (quiescent RGC)”?
Line 151. The subtopic should be numbered “3.1” instead of “1.1”.
Line 158. Please revise all abbreviations used throughout the text. “Neural stem cell” has been first abbreviated as “NSC” in line 31, but the abbreviation is not used consistently (e.g. see line 188). Once an abbreviation is introduced, it should be used throughout the paper.
Line 172. “At their target location they differentiate into mature neurons expressing well-characterized neuronal markers (i.e. HuC/D, Pax6a, PV) and display signs of functional integration such as synaptogenesis”. Are RGCs or Type 3 cells able to differentiate into glial cells as well? If so, is it affected by brain injury?
Line 178. Please capitalize all letters in “Gfap”.
Line 191. “The (+/-) means that these markers are expressed at lower levels in the subtype.”. What does (-) mean in table 1?
Line 195. The subtopic should be numbered “3.2” instead of “1.2”.
Line 212. What does “TH” stand for? (tyrosine hydroxylase??) It has not been mentioned before.
Line 220. “Mechanical injury of the telencephalon by either inserting a small cannula through the skull or through the nasal cavity remains the most investigated model in zebrafish for studying brain regeneration”. What is the difference between the two procedures in terms of brain damage? Isn’t the one performed through the nasal cavity more susceptible to damaging the olfactory organ and/or affecting olfactory neurogenesis? This should be highlighted and discussed.
Page 9 (Figure 3). It should be mentioned somewhere in the legend that the cellular response on the contralateral side is not shown. Otherwise, it seems that there is no contralateral response after brain injury. Also, it would be helpful to discriminate which cellular response is shown in each row of Figure 3 in zebrafish and rodents (ex., In zebrafish, the first row shows cell death and microglial response).
Line 230. What do “hpl” and “dpl” stand for? (hours/days post lesion??) Please define this in the legend of Figure 3 and the text.
Line 271. Please add the scale bar measurement to Figure 4. Arrows were not defined either.
Line 294. Which inflammatory molecules (pro, anti, or both) are those? Examples?
Line 296. This is an interesting piece of information but it’s missing some details, such as, which neurogenic event was impaired by inhibition of microglial activation? Which molecular pathway is involved in this process? Is it related to one of those molecules mentioned in the previous paragraph (interleukins or TNF, TGF)?
Line 310. “They observed an increased number of OPCs and mature oligodendrocytes at 1 dpl (in 50% of the studied brains). This accumulation of OPCs is more prominent between 2 dpl and 14 dpl and is detected in almost all the studied brains (94%).” This accumulation of OPC between 2 and 14 dpl is not shown in Figure 3.
Line 355. Consider naming this topic as “Characterization of neurogenic niches in the telencephalon of adult mammals” instead of “Characterization of neurogenic niches in the telencephalon of adult rodents”, because this topic includes characterization of neurogenic niche in humans as well.
Line 423. It’s not clear why the authors are comparing TBI models in zebrafish and stroke models in rodents. The argument of the number of publications is too vague. What kind of information is missing in the current literature (7.847 papers) about TBI that made the authors focus on stroke? Cellular and molecular mechanisms? Any specific inflammatory response? Also, even though, “TBI and stroke lead to almost similar cellular events and disorders…” – as stated by the authors - it is hard to compare those models in terms of the inflammatory response, especially to track cellular events over time, which was one of the topics explored in this review. According to Bramlett and Dietrich (2004) – already cited in the reference list - “the fact that these injuries arise from different types of primary insults leads to diverse cellular vulnerability patterns as well as a spectrum of injury processes. Blunt head trauma produces shear forces that result in primary membrane damage to neuronal cell bodies, white matter structures, and vascular beds as well as secondary injury mechanisms. Severe cerebral ischemic insults lead to metabolic stress, ionic perturbations, and a complex cascade of biochemical and molecular events ultimately causing neuronal death.” This issue deserves a better approach/attention in the text, as it is not convincing that these two mechanisms of damage result in comparable outcomes.
Line 469. According to Lourbopoulos et al., 2015 (cited by the authors), in mammals, microglia are extremely plastic and respond with a range of phenotypes depending on the brain homeostasis status or resulting from pathology, including during stroke, where they react depending on the stroke regions (infart core and pre-infart core). These microglial phenotypes should be highlighted and discussed in this section. The way it was described here, it seems a “general microglia response” with no specificity. How do microglial phenotypes contribute to brain repair and adult neurogenesis in mammals? (See also Willis et al., 2020)
Lines 469 and 260. Wu et al., 2020 show two phenotypically distinct microglial populations in adult zebrafish; they differ not only in morphology, but in distribution, development, and function. How might these microglial phenotypes be comparable with those phenotypes in mammals? How would it contribute to this high regenerative capacity of brain repair in zebrafish?
Line 572. Which neurogenic event is induced in TBI models? Proliferation, differentiation…? Any particular cell type (C-cell, A-cell) is more responsive?
Line 576. It is not clear what those signs mean (+/-, +, -, +++). Please define them.
Line 577 (table 2). Considering that time is a key component related to glial response in brain repair, it would be helpful to indicate the time of each event in this table. For example, microglial recruitment occurs in both zebrafish and mammals, but it persists in a different time frame.
Line 599. It needs to be mentioned in this section the dual roles of astrocytes in glial scar formation. According to Zhou et al., 2020 (cited by the authors) “selective stimulation of the beneficial astrocyte derived molecules and simultaneous attenuation of the deleterious factors based on the spatiotemporal-environment can provide a promising astrocyte-targeting therapeutic strategy”. How would “beneficial astrocytes” contribute to neuronal plasticity and neurogenesis in stroke/TBI? It is also relevant to briefly explain how glial scar might have a positive effect on brain injury as well (for example, acting as a barrier isolating the damaged area or containing the spread of inflammatory cells). This dual role is not clear in the text.
Line 604. Briefly, how do extracellular matrix components from astrocytes effectively impair brain regeneration? How are they related to the inflammatory response? Possible inflammatory factors involved?
Line 637. This conclusion is too long! Most of what was written in there would fit perfectly in the previous topic 6.6: “Glial scar: a paradigm for understanding the difference between zebrafish and mammalian regeneration?”. The conclusion has a great paragraph (line 659) explaining how glial scar might hamper NSCs access to injury areas in both zebrafish and mammals. The same thought applies to other paragraphs in the conclusion (line 677 – about cell death) that would fit nicely in the cell death topic.
Line 652. “…zebrafish appear to respond to injury by turning on genes that are not activated in other models and to additionally activate epigenetic programs.” What kind of genes?
Reference:
Willis et al., 2020. Repopulating Microglia Promote Brain Repair in an IL-6-Dependent Manner. Cell. https://doi.org/10.1016/j.cell.2020.02.013
Author Response
Response to Reviewer#1
This review presents two interesting topics on brain repair: adult neurogenesis and cellular mechanisms triggered by injury, focusing both on similarities and differences in these processes in adult zebrafish and mammals. The authors selected an impressive amount of impactful literature in the area and outlined important aspects related to brain repair in evolutionary and physiological contexts. However, its presentation suffers from having a confusing structural organization, as well as being mostly descriptive with less thoughtful discussion/comparison between zebrafish and mammals. Providing more details of the material across species would significantly enhance the value of this review.
We thank the Reviewer for these comments. As suggested by the Reviewer, we significantly rearranged our manuscript by combining point 2 with point 5 and by adding the part about regenerative events of mammals to the zebrafish section. This allows a better comparison between zebrafish and mammals, considering the neurogenic niches and the cellular and anatomical characterization of the respective niches.
General comments
The abbreviations need to be carefully reviewed all over the manuscript.
According to the Reviewer’s suggestion, we carefully checked all abbreviations and made changes where they were necessary.
The structural organization of the manuscript is a bit confusing. In the abstract, the authors state that they “will first discuss the similarities and differences regarding (i) the distribution of neurogenic niches in the brain of adult zebrafish and mammals (mainly mouse)” but this was not clear until page 12-13 when they finally describe the characterization of rodent neurogenic niche and then show the similarities and differences. The same happens with other topics, such as brain damage and brain repair. Rather than presenting this information in this linear fashion, it would be more interesting and helpful for the reader to integrate the content related to the different animal models.
We understood the concerns of the Reviewer and rearranged these two parts accordingly.
It’s not clear why authors focused on TBI models in zebrafish and stroke models in rodents to compare cellular events after brain injury. Although “TBI and stroke lead to almost similar cellular events and disorders – as stated by the authors -, these injuries have different types of primary insult and then different cellular vulnerability, which becomes a challenge to study timeline events related to inflammatory responses. It would be more interesting to focus on TBI models in rodents as well and use the stroke data (as a secondary topic) to reinforce the main idea on brain injury.
We understood the comments and concerns of Reviewer and updated the explanation for this concern. Furthermore, stroke and TBI result in similar recovery processed without striking differences in cognitive performance suggesting similar regenerative outcomes (Castor 2018).
We also added a table (new Table 2) that provides the similar features of TBI and stroke demonstrating that most of processes occurring during stroke. For instance, there is also a decreased of blood flow in stroke. Consequently, although differences exist between stroke and TBI, they mostly results in common consequences and outcomes.
The discussion is slightly superficial. Some topics are missing detail and focus.
We do not agree with this comment of reviewer#1. However, the discussion was slightly changed in order to further answer to the requirements of Reviewer.
Specific comments
Line 18 (Abstract). “we will first discuss the similarities and differences regarding (i) the distribution of neurogenic niches in the brain of adult zebrafish and mammals (mainly mouse)”. These similarities and differences are not clear at the beginning of the manuscript, but instead wait several pages for the comparison information to appear. It would be far more helpful for the reader if the authors could use their expertise to integrate this information and compare these models from the beginning, as is suggested by the abstract.
According to the Reviewer’s suggestions, we merged the explanations of the neurogenic niches in zebrafish, rodents and humans into one part (point 2).
Page 4 (Figure 1). This is a great figure that brings together a nice comparison of neurogenic niches throughout the brain in the three groups (zebrafish, rodents, and humans) but the discussion of similarities and differences is spread over the text. For example, page 2 (line 77) describes the neurogenic niche in zebrafish but the rodent’s characterization is only described on page 12 (line 355). Why not put this information together? The same happens in Figure 3.
Again, we agree with the Reviewer and put the explanations all in one part to make it more clearly structured.
Line 105. Please check the legend of Figure 1; it is missing a couple of abbreviations, such as RGC, Dp, Cx, and RMS. Also, consider adding letters to Figure 1 (ex., E, F, G…) to identify rodent and human brain sections.
We thank Reviewer #1 for this comment. We added the missing abbreviations and completed the legends accordingly.
Line 107. “The mammalian brain displays only two main neurogenic niches: the subventricular zone (SVZ) of the lateral ventricles and subgranular zone of the dentate gyrus (DG) of the hippocampus.” Figure 1 shows a neurogenic area in the hypothalamus (Figure 1A) of rodents as well; it would be helpful to mention that in the legend or consider removing it.
We agree with Reviewer #1. The two main neurogenic niches of the adult mammalian brain are the SGZ of the DG and SVZ but some discrete neurogenic niches have been described more recently, such as the hypothalamus. We consequently added a sentence in the legend: “Note that the mammalian hypothalamus (HYP) also exhibits discrete neurogenesis.”
Figure 1, 2, and line 141. The nomenclature of RGC cells is not consistent. Is there any difference between “Non proliferative RGCs (Type1)” or “quiescent (Type 1)” or “Type 1 (quiescent RGC)”?
There is no difference: Type 1 cells in zebrafish correspond to non-proliferative RGCs, which are quiescent cells. The legend of figure 1 was modified as follows: Type 1 (non proliferative/quiescent RGC)
Line 151. The subtopic should be numbered “3.1” instead of “1.1”.
We thank Reviewer #1 for bringing this to our attention. We changed the numbering throughout the text and incorporated the rearrangements of the different paragraphs.
Line 158. Please revise all abbreviations used throughout the text. “Neural stem cell” has been first abbreviated as “NSC” in line 31, but the abbreviation is not used consistently (e.g. see line 188). Once an abbreviation is introduced, it should be used throughout the paper.
According to the Reviewer’s suggestion we revised the manuscript and made consistent changes for NSCs, RMS, SGZ, SVZ, DG and TBI.
Line 172. “At their target location they differentiate into mature neurons expressing well-characterized neuronal markers (i.e. HuC/D, Pax6a, PV) and display signs of functional integration such as synaptogenesis”. Are RGCs or Type 3 cells able to differentiate into glial cells as well? If so, is it affected by brain injury?
According to the zebrafish literature, RGCs can self-renew and/or generate new neurons directly or through intermediate progenitors (type 3 cells). Until now, no astrocytes have been observed in the telencephalic regions and neither RGCs or type 3 cells have been shown to generate astrocytic, oligodendrocyte or microglial cells in the brain. However, in the zebrafish spinal cord, olig2+ RGCs have been shown to continuously generate new oligodendrocytes.
After brain injury, a shift in the mode of division of NSCs has been suggested by Barbosa et al., 2015 when compared to the brain (symmetric gliogenic division versus symmetric non-gliogenic division). Additionally, new oligodendrocytes have been suggested to be generated from telencephalic olig2+ RGCs following injury. This new paper, published after the submission of this review, is now quoted (Kim et al 2020).
In brief, under homeostatic conditions, RGCs self-renew and neurons directly or indirectly through type 3 cells.
We added the following sentences to the appropriate paragraph:
- “Interestingly, compared to constitutive neurogenesis, a shift in the mode of division of NSCs has been observed by Barbosa and colleagues during regenerative neurogenesis (Barbosa et al., 2015), that could lead to a depletion of NSC.”
- “However, recent data suggest a higher number of proliferating parenchymal and ventricular olig2-positive cells at 4 dpl (Kim et al., 2020). This study also suggests that olig2-positive RGCs from the medial telencephalic ventricular zone can generate new oligodendrocytes following brain injury (Kim et al., 2020).”
Line 178. Please capitalize all letters in “Gfap”.
We understand the comment of Reviewer #1. However, according to the zebrafish nomenclature, the protein name should be written as Gfap and not GFAP. According to this nomenclature, the notation of all other zebrafish proteins has been revised throughout the manuscript (i.e. Blbp, Pcna…).
Note the differences between zebrafish and mammalian naming conventions:
species / gene / protein
zebrafish /shha/ Shha
Line 191. “The (+/-) means that these markers are expressed at lower levels in the subtype.”. What does (-) mean in table 1?
We thank Reviewer#1 for this comment. As (-) means that the marker is not expressed, it is not useful to mention it, and we consequently deleted Gfap (-) from the table.
Line 195. The subtopic should be numbered “3.2” instead of “1.2”.
The numbering has been changed throughout the text, given the rearrangements of the different paragraphs.
Line 212. What does “TH” stand for? (tyrosine hydroxylase??) It has not been mentioned before.
We again thank the Reviewer for bringing this up and replaced TH by Tyroxine Hydroxylase
Line 220. “Mechanical injury of the telencephalon by either inserting a small cannula through the skull or through the nasal cavity remains the most investigated model in zebrafish for studying brain regeneration”. What is the difference between the two procedures in terms of brain damage? Isn’t the one performed through the nasal cavity more susceptible to damaging the olfactory organ and/or affecting olfactory neurogenesis? This should be highlighted and discussed.
Indeed Reviewer #1 is correct in his assumption that in the 2nd model, the needle can go through the OB (but not necessarily). However, both models lead to almost similar cell reactivity and there are few data documenting the potential differences in olfactory neurogenesis induced by the two different models. To explain this circumstance, the following sentence was added to the main text: “Interestingly, the second injury method could lead to a damage of the olfactory bulbs but does not alter the brain repair mechanisms substantially. »
Page 9 (Figure 3). It should be mentioned somewhere in the legend that the cellular response on the contralateral side is not shown. Otherwise, it seems that there is no contralateral response after brain injury. Also, it would be helpful to discriminate which cellular response is shown in each row of Figure 3 in zebrafish and rodents (ex., In zebrafish, the first row shows cell death and microglial response).
We followed the Reviewer’s suggestion and modified the legends accordingly.
Line 230. What do “hpl” and “dpl” stand for? (hours/days post lesion??) Please define this in the legend of Figure 3 and the text.
The reviewer is correct: “hpl” stands for “hours post lesion” and “dpl” for “days post lesion”. The abbreviations have now been defined in the legend of figure 3 and in the text.
Line 271. Please add the scale bar measurement to Figure 4. Arrows were not defined either.
In accordance with the Reviewer’s suggestion, we added a scale bar to figure 4 and explained in the figure legend what the arrows represent.
Line 294. Which inflammatory molecules (pro, anti, or both) are those? Examples?
The text has been modified as follows: “As in mammals, pro-inflammatory molecules (i.e. Interleukins Il-1β, Il-6, Il-8 and TNFα) secreted…”
Line 296. This is an interesting piece of information but it’s missing some details, such as, which neurogenic event was impaired by inhibition of microglial activation? Which molecular pathway is involved in this process? Is it related to one of those molecules mentioned in the previous paragraph (interleukins or TNF, TGF)?
For more clarification we added the following sentences: “For instance, the inhibition of microglia activation during zebrafish brain injury leads to a decreased expression of Tnf-α and phospho-stat3/β-catenin signaling, which results in a lower proliferation of neural progenitor/stem cells and a lower number of newborn neurons without affecting differentiation (Kanagaraj et al., 2020). These new data decipher the key roles of microglia in brain repair mechanisms.”
Line 310. “They observed an increased number of OPCs and mature oligodendrocytes at 1 dpl (in 50% of the studied brains). This accumulation of OPCs is more prominent between 2 dpl and 14 dpl and is detected in almost all the studied brains (94%).” This accumulation of OPC between 2 and 14 dpl is not shown in Figure 3.
We are grateful to Reviewer#1 for highlighting this mistake.
Line 355. Consider naming this topic as “Characterization of neurogenic niches in the telencephalon of adult mammals” instead of “Characterization of neurogenic niches in the telencephalon of adult rodents”, because this topic includes characterization of neurogenic niche in humans as well.
We made that change, as suggested. We also fused this section with point 2 for more clarity
Line 423. It’s not clear why the authors are comparing TBI models in zebrafish and stroke models in rodents. The argument of the number of publications is too vague. What kind of information is missing in the current literature (7.847 papers) about TBI that made the authors focus on stroke? Cellular and molecular mechanisms? Any specific inflammatory response?
Also, even though, “TBI and stroke lead to almost similar cellular events and disorders…” – as stated by the authors - it is hard to compare those models in terms of the inflammatory response, especially to track cellular events over time, which was one of the topics explored in this review. According to Bramlett and Dietrich (2004) – already cited in the reference list - “the fact that these injuries arise from different types of primary insults leads to diverse cellular vulnerability patterns as well as a spectrum of injury processes. Blunt head trauma produces shear forces that result in primary membrane damage to neuronal cell bodies, white matter structures, and vascular beds as well as secondary injury mechanisms. Severe cerebral ischemic insults lead to metabolic stress, ionic perturbations, and a complex cascade of biochemical and molecular events ultimately causing neuronal death.” This issue deserves a better approach/attention in the text, as it is not convincing that these two mechanisms of damage result in comparable outcomes.
We understand this concern. As explained, we tried to compare the most studied models of brain injury in fish and mammals. The different types of TBI protocols used (FPI, CCI, PBBI), the reproducibility and standardization makes the results of TBI complicated to analyze. The literature is much more abundant for stroke and even if there are differences to the stroke protocol, some studies have performed integrative work looking at microglia, astrocyte, oligodendrocyte and proliferative activation.
Indeed, as mentioned by reviewer #1: “Severe cerebral ischemic insults lead to metabolic stress, ionic perturbations, and a complex cascade of biochemical and molecular events ultimately causing neuronal death.” However, these features are also shared in TBI as reviewed by Bramlett and Dietrich. Furthermore, as discussed in the same review concerning the blood flow, “mild to moderate TBI in experimental animals, reductions in flow are commonly 70–80% of normal (Dietrich et al., 1996a). However, with more severe injury, CBF values approach ischemic levels (Dietrich et al., 1998a).”
Taken together, these data demonstrate that even if differences exist between stroke and TBI, the main mechanisms are involved.
In order to better discuss TBI and stroke, the TBI part has been updated throughout the text and a table has been added showing the similarities between stroke and TBI.
Finally, recent studies in humans show that TBI and stroke did not result in differences in recovery (Castor 2018), suggesting that insults, their consequences and subsequent brain repair do not strongly differ.
Line 469. According to Lourbopoulos et al., 2015 (cited by the authors), in mammals, microglia are extremely plastic and respond with a range of phenotypes depending on the brain homeostasis status or resulting from pathology, including during stroke, where they react depending on the stroke regions (infart core and pre-infart core). These microglial phenotypes should be highlighted and discussed in this section. The way it was described here, it seems a “general microglia response” with no specificity. How do microglial phenotypes contribute to brain repair and adult neurogenesis in mammals? (See also Willis et al., 2020).
We modified this part and discuss a bit more the phenotype and the subtypes of microglia. “Importantly, microglial cells display a huge diversity of phenotypes and reactivity allowing extreme plasticity. Microglia show different regional density (being more dense in the telencephalon and in myelinated regions), can be differently activated following injury (even in regions for which no neuronal death occurs) and display morphological change with age (Olah et al., 2011). As for macrophages, microglia can exhibit M1 (pro-inflammatory) or M2 (anti-inflammatory) phenotypes, the M2 phenotype being more associated with resolution of inflammation/regeneration processes, namely in animal models of stroke (Olah et al., 2011; Michell-Robinson et al., 2015; Orihuela et al., 2015). In order to add complexity, it was also strongly suggested that microglia harbor different subtypes (at least 6) in the brain of mammals endowed with peculiar genomic, spatial, morphological, and functional specializations (Dubbelaar et al., 2018; Stratoulias et al., 2019). Microglia have also been shown to be part of the neurogenic niches. They can have positive or negative effects on neurogenesis according to their activation and secreted molecules. Thus, ischemia and cell death could initiate IGF-1 and TGF ß expression and promote neurogenesis (Olah et al., 2011). Very interestingly, new data strongly suggest that a resident microglial population does not inhibit endogenous brain regeneration processes in mouse following TBI, but rather cannot support these processes (Willis et al., 2020). A pro-regenerative phenotype can also be induced in mammals through IL-6 trans signaling (Willis et al., 2020), demonstrating that inflammatory signals are important contributors to brain repair mechanisms, as in regenerative organisms like zebrafish.”
Lines 469 and 260. Wu et al., 2020 show two phenotypically distinct microglial populations in adult zebrafish; they differ not only in morphology, but in distribution, development, and function. How might these microglial phenotypes be comparable with those phenotypes in mammals? How would it contribute to this high regenerative capacity of brain repair in zebrafish?
We understand the reviewer’s question and refer to our explanation under the previous point.
Line 572. Which neurogenic event is induced in TBI models? Proliferation, differentiation…? Any particular cell type (C-cell, A-cell) is more responsive?
We modified this part as follows “Thus, TBI has been shown to promote the reactivation of quiescent NSCs that actively divide producing new neural progenitors (Wang et al., 2016). The newly generated neuroblasts will migrate in chain to the lesioned areas (Chang et al., 2016). As reviewed by Chang et al (2016), the different models of TBI seem to globally lead to increased NSC proliferation, migration and differentiation but a wide heterogeneity in the TBI responses is observed probably due to the differences between the severity, location, timing and types of injury.”
Line 576. It is not clear what those signs mean (+/-, +, -, +++). Please define them.
We thank reviewer #1 for pointing this out and defined the meaning in the legend.
Line 577 (table 2). Considering that time is a key component related to glial response in brain repair, it would be helpful to indicate the time of each event in this table. For example, microglial recruitment occurs in both zebrafish and mammals, but it persists in a different time frame.
We agree with Reviewer #1. However, the main idea of this table is to show what is common and different between fish and mammals considering the main points of brain damage. Introducing time would complexify the table and will not allow a clear overview of what is shared or not.
Line 599. It needs to be mentioned in this section the dual roles of astrocytes in glial scar formation. According to Zhou et al., 2020 (cited by the authors) “selective stimulation of the beneficial astrocyte derived molecules and simultaneous attenuation of the deleterious factors based on the spatiotemporal-environment can provide a promising astrocyte-targeting therapeutic strategy”. How would “beneficial astrocytes” contribute to neuronal plasticity and neurogenesis in stroke/TBI? It is also relevant to briefly explain how glial scar might have a positive effect on brain injury as well (for example, acting as a barrier isolating the damaged area or containing the spread of inflammatory cells). This dual role is not clear in the text.
We agree with Reviewer 1 and this is now done.
“Astrocytes appear as key cell in the development of the glial scar that is essential in the establishment of a physical and chemical barrier that isolates the damaged area and contains the spread of inflammatory cells. Inhibiting or promoting astrogliosis (and so glial scar) did not have striking curative effects [160]. However, the selective stimulation of beneficial astrocyte-derived molecules could represent an interesting therapeutic way to promote blood-brain barrier repair, neurogenesis and synaptic plasticity [160].
Line 604. Briefly, how do extracellular matrix components from astrocytes effectively impair brain regeneration? How are they related to the inflammatory response? Possible inflammatory factors involved?
We tried to answer this question of Reviewer #1 by adding the following sentence to the text: “In addition to representing a physical barrier, these extracellular matrix molecules can also lead to growth cone collapse, axonal guidance inhibition, as well as neural progenitor migration defects through activation of RhoA/ROCK signaling (Galindo et al., 2018). »
Line 637. This conclusion is too long! Most of what was written in there would fit perfectly in the previous topic 6.6: “Glial scar: a paradigm for understanding the difference between zebrafish and mammalian regeneration?”. The conclusion has a great paragraph (line 659) explaining how glial scar might hamper NSCs access to injury areas in both zebrafish and mammals. The same thought applies to other paragraphs in the conclusion (line 677 – about cell death) that would fit nicely in the cell death topic.
Along with the reviewer’s suggestions, the conclusion has been shortened in some parts, to be more focused and developed concerning the potential role of cell death and microglial cells in brain repair.
Line 652. “…zebrafish appear to respond to injury by turning on genes that are not activated in other models and to additionally activate epigenetic programs.” What kind of genes?
We changed this sentence to “Among them, zebrafish appear to respond to brain damage by turning on genes such as gata3 and interleukin-4 receptor, that are not activated in rodent models (Mashkaryan et al., 2020; Yoshiya et al., 2003; Chang et al., 2016,Celikkaya et al., 2019). Additionally, changes in regeneration-responsive enhancers of mammals might be one of the reasons for less regenerative capacities in mammals, when compared to zebrafish. This was recently suggested for the inhibin beta A gene (Wei Wang,2020). »
Reference:
Willis et al., 2020. Repopulating Microglia Promote Brain Repair in an IL-6-Dependent Manner. Cell. https://doi.org/10.1016/j.cell.2020.02.013
Reviewer 2 Report
Major Comments:
The authors give a comprehensive review of the literature related to cellular changes that occur in response to brain injury. They focus mainly on related research using the teleost fish, but also discuss research in rodent models for comparison. To me, the new insights in this manuscript come from the comparison between fish and rodent responses to brain injury, which is nicely integrated in the figures. However, this comparison is segregated in the text, making it difficult to follow. I would suggest the authors consider merging the rodent text sections with the fish sections. For example, combining Section 2, Location of neurogenic niches in the brain of adult zebrafish, with Section 5, Characterization of neurogenic niches in the telencephalon of adult rodents. I think the main point to compare the injury response and regenerative properties of fish and rodents will be clearer if this formatting change is done throughout the manuscript. Additionally, the text will match the flow of the figures.
Minor Comments:
- There are multiple typos of Greek letters replaced by strange symbols in:
Page 7, line 179
Page 10, line 277
Page 17, line 614
- Page 12, line 377: “granular neurons” should be changed to “dentate granule neurons”
- Page 12, line 378: “dentate gyrus” should be changed to “DG” because it was already defined earlier
Author Response
Response to Reviewer#2
The authors give a comprehensive review of the literature related to cellular changes that occur in response to brain injury. They focus mainly on related research using the teleost fish, but also discuss research in rodent models for comparison. To me, the new insights in this manuscript come from the comparison between fish and rodent responses to brain injury, which is nicely integrated in the figures. However, this comparison is segregated in the text, making it difficult to follow. I would suggest the authors consider merging the rodent text sections with the fish sections. For example, combining Section 2, Location of neurogenic niches in the brain of adult zebrafish, with Section 5, Characterization of neurogenic niches in the telencephalon of adult rodents. I think the main point to compare the injury response and regenerative properties of fish and rodents will be clearer if this formatting change is done throughout the manuscript. Additionally, the text will match the flow of the figures.
We are very grateful for the comments of Reviewer #2. Accordingly, we merged the explanations of the neurogenic niches in zebrafish, rodents and humans into one part (point 2).
Minor Comments:
There are multiple typos of Greek letters replaced by strange symbols in:
Page 7, line 179
Page 10, line 277
Page 17, line 614
We thank Reviewer #2 for bringing this to our attention and fixed the letters.
Page 12, line 377: “granular neurons” should be changed to “dentate granule neurons”
According to the Reviewer’s suggestions, we changed “granular neurons” to “dentate granule neurons” in this sentence.
Page 12, line 378: “dentate gyrus” should be changed to “DG” because it was already defined earlier
Along with the reviewer’s suggestion we changed it to DG.
Reviewer 3 Report
The review is very clear, well-written and raises the most important issues on the subject, which is adult neurogenesis. I have only a few minor comments:
Page 5 - Caption for figure 1 (Line 114). Please insert “(RGC)” after “...are quiescent and proliferative radial glial cells...” to explain the abbreviation used in the Figure.
Page 6 - Caption for Figure 2. Again, please, explain the abbreviation RGC.
Page 7 - Line 178. Please use GFAP instead of Gfap since in all other times that the abbreviation of this marker appears it is written GFAP.
Page 8 - Line 235. The abreviation “dpl” was not defined. I imagine that it is “day post-lesion) but I am not sure! The same for “hpl” on line 230, same page (is it hours post-lesion?).
Figure 3 - Please turn horizontally (mirror-like) all cells depicted in the scheme legend. I also suggest to increase the size of these cells in order to turn them more visible to the reader.
Page 14 - Line 430. BBB (blood-brain barrier) should be defined.
Page 16 - Line 552. Please define KO as knockout.
Page 17 - Line 609. You have already used BBB, although it was not defined the first time it appeared in the text on page 14, line 430.
Conclusions – This part is clear and sumarizes the state-of-the-art on the subject; however, since it raises an evolutionary aspect, it would be very interesting at this point, to consider adding some information on higher invertebrates. There are studies on molecular and cellular mechanisms involved in the adult neurogenesis of many invertebrate species - in neurogenic niches and migratory routes -, and they also show evidences for an association between vascular cells and the niche precursors. They support that cells of the blood lineage are not only associated with the roles that are generally attributed to them, but are the cells that either signal other cell types to differentiate into neural cells.
Author Response
Reviewer#3
The review is very clear, well-written and raises the most important issues on the subject, which is adult neurogenesis. I have only a few minor comments:
Page 5 - Caption for figure 1 (Line 114). Please insert “(RGC)” after “...are quiescent and proliferative radial glial cells...” to explain the abbreviation used in the Figure.
We made this change as suggested
Page 6 - Caption for Figure 2. Again, please, explain the abbreviation RGC.
Along with the Reviewers suggestion, we explained the abbreviation
Page 7 - Line 178. Please use GFAP instead of Gfap since in all other times that the abbreviation of this marker appears it is written GFAP.
We understand the comments from Reviewer #3 and are thankful for directing our attention to this inconsistency. However, according to the zebrafish nomenclature, the protein name should be written as Gfap. We changed it to Gfap throughout the entire manuscript.
Note the differences between zebrafish and mammalian naming conventions:
species / gene / protein
zebrafish /shha/ Shha
Page 8 - Line 235. The abreviation “dpl” was not defined. I imagine that it is “day post-lesion) but I am not sure! The same for “hpl” on line 230, same page (is it hours post-lesion?).
Reviewer #3 is right. Indeed” dpl” stands for “days post lesion”, and “hpl” for “hours post lesion”. We made this correction on page 8.
Figure 3 - Please turn horizontally (mirror-like) all cells depicted in the scheme legend. I also suggest to increase the size of these cells in order to turn them more visible to the reader.
We are not sure to understand what Reviewer asked for, but we provided a modified figure.
Page 14 - Line 430. BBB (blood-brain barrier) should be defined.
We made the change, according to Reviewer #3’s suggestion.
Page 16 - Line 552. Please define KO as knockout.
We made the change.
Page 17 - Line 609. You have already used BBB, although it was not defined the first time it appeared in the text on page 14, line 430.
We changed “BBB” to “blood-brain barrier” throughout the entire manuscript.
Conclusions – This part is clear and sumarizes the state-of-the-art on the subject; however, since it raises an evolutionary aspect, it would be very interesting at this point, to consider adding some information on higher invertebrates. There are studies on molecular and cellular mechanisms involved in the adult neurogenesis of many invertebrate species - in neurogenic niches and migratory routes -, and they also show evidences for an association between vascular cells and the niche precursors. They support that cells of the blood lineage are not only associated with the roles that are generally attributed to them, but are the cells that either signal other cell types to differentiate into neural cells.
We agree with Reviewer. However, this is not the main aim of the review and another reviewer complain about the discussion and its length. We nevertheless modified the last part of the discussion by adding this:
« Interestingly, studies on invertebrates such as drosophila also demonstrate common strategies in neurogenesis and brain repair and highlight the role of blood vessels in these mechanisms [186,187]. Thus, the study of different taxa is of great interest in order to better understand neurogenesis and brain repair through evolutionary conserved processes.